# Structure-Guided Strategies of Targeted Therapies for Patients with *EGFR*-Mutant Non–Small Cell Lung Cancer

**DOI:** 10.3390/biom13020210

**Published:** 2023-01-20

**Authors:** Zhenfang Du, Jinghan Sun, Yunkai Zhang, Nigaerayi Hesilaiti, Qi Xia, Heqing Cui, Na Fan, Xiaofang Xu

**Affiliations:** 1Department of Genetic and Developmental Biology, School of Medicine, Southeast University, Nanjing 210003, China; 2School of Life Science and Technology, Southeast University, Nanjing 210018, China; 3BioSpatula LLC, Telford, PA 18969, USA; 4Department of Radiotherapy, Nanjing Chest Hospital, Nanjing Medical University Affiliated Brain Hospital, Nanjing 210029, China; 5Department of Respiratory Medicine and Critical Care Medicine, The Second Affiliated Hospital of Xi’an Jiaotong University, Xi’an 710004, China; 6Department of Thoracic Surgery, The Cancer Hospital of the University of Chinese Academy of Sciences (Zhejiang Cancer Hospital), Institute of Basic Medicine and Cancer (IBMC), Chinese Academy of Sciences, Hangzhou 310022, China

**Keywords:** non–small cell lung cancer, EGFR, structure, targeted therapy

## Abstract

Oncogenic mutations within the EGFR kinase domain are well-established driver mutations in non–small cell lung cancer (NSCLC). Small-molecule tyrosine kinase inhibitors (TKIs) specifically targeting these mutations have improved treatment outcomes for patients with this subtype of NSCLC. The selectivity of these targeted agents is based on the location of the mutations within the exons of the *EGFR* gene, and grouping mutations based on structural similarities has proved a useful tool for conceptualizing the heterogeneity of TKI response. Structure-based analysis of *EGFR* mutations has influenced TKI development, and improved structural understanding will inform continued therapeutic development and further improve patient outcomes. In this review, we summarize recent progress on targeted therapy strategies for patients with *EGFR*-mutant NSCLC based on structure and function analysis.

## 1. Introduction

Lung cancer is the most common cause of cancer-related death worldwide, accounting for 21.5% of male and 13.7% of female deaths from cancer in 2020 [1]. Histologically, 85% of patients with lung cancer can be classified with non–small cell lung cancer (NSCLC). Mutations within the epidermal growth factor receptor (EGFR) kinase domain are the root cause of a subtype of NSCLC and account for up to 50% of NSCLC, depending on demography [2,3,4,5], with higher proportions observed in females (42% of diagnoses) [6], east Asians (47% of diagnoses) [4], and nonsmokers (30.6% of diagnoses) [7]. The administration of small-molecule tyrosine kinase inhibitors (TKIs) specifically targeting mutant EGFR has made considerable progress in the treatment of this subset of patients compared with conventional chemotherapy [8,9,10,11].

In normal human physiology, the epidermal growth factor ligand binds to the extracellular domain of EGFR and induces conformational alterations [12,13,14]. These alterations release the autoinhibition of the basal state [15] and drive the asymmetric dimerization of intracellular kinase domains, in which one kinase domain acts as activator (“donor”) and the other as receiver (“acceptor”). The kinase domains of the asymmetric dimer phosphorylate the C-terminals of each other in a *trans* manner and transmit the signal from extracellular to intracellular [16]. Asymmetric dimers finally undergo multimerization, or higher-order oligomerization, in the manner of a “head-to-tail” chain [17,18]. Different ligands bind to EGFR and induce varying receptor conformations, which subsequently leads to differential downstream signaling [19]. Oncogenic mutations within the EGFR kinase domain lead to constitutive activation of the receptor and its downstream signaling, which promotes cell survival and proliferation [20,21,22,23,24]. Small-molecule TKIs reversibly or irreversibly bind the ATP pocket of the EGFR kinase domain and inhibit receptor autophosphorylation and downstream signaling [20,25].

TKI sensitivity can be classified based on the distribution of mutations within the exons of the *EGFR* gene [26]. The majority of *EGFR* mutations are classical mutations, including the canonical exon 19 deletions (Ex19Del) and the L858R point mutation in exon 21, which together account for up to 90% of all *EGFR* mutations and confer sensitivity to EGFR TKIs [27]. (Note: In this review, we use the most common EGFR protein nomenclature, which includes numbering a 24-residue sequence found in immature EGFR; thus, for example, we will use L858R instead of L834R). The remaining *EGFR* mutations (those that are not Ex19Del or L858R) are uncommon or rare mutations with variable TKI sensitivity [28]. Mutations within exons 18, 19 and 21 are sensitive (e.g., G719X, L747P and L861Q, correspondingly) [29,30,31,32,33,34,35,36], while most exon 20 mutations are insensitive to EGFR TKIs (e.g., most exon 20 insertions) [37,38,39,40,41]. Exceptionally, S768I is an exon 20 mutation sensitive to first- and third-generation EGFR TKIs [42], though less so than the canonical L858R mutation [43]. A763_Y764insFQEA, one of the exon 20 insertions, is located in the middle of the αC helix. The FQEA insertion shifts the register of the αC helix toward its N terminus, resulting in a longer β3–αC loop and activating the receptor by destabilizing the inactive conformation in a related manner to the classical mutation L858R [21,44]. Therefore, differently from other exon 20 insertions, A763_Y764insFQEA confers sensitivity to first-, second- and third-generation EGFR TKIs [45]. The discordance within these traditional exon-based groupings demonstrates the shortcomings of this conceptual framework.

Structural studies have improved our understandings of EGFR biology and TKI sensitivity, and are a powerful tool in EGFR TKI drug discovery [27,46,47,48]. Using hierarchical clustering of TKI selectivity and mutational mapping, Robichaux et al. classified *EGFR* kinase domain mutations into four distinct subtypes: (1) classical-like mutations that are far from the ATP-binding pocket; (2) T790M-like mutations that are located in the hydrophobic core; (3) exon 20 insertions in the αC–β4 Loop following the C-terminal end of the αC-helix (Ex20ins-L); (4) P-loop and αC-helix compressing (PACC) mutations that are located on the interior surface of the ATP-binding pocket or C-terminal portion of the αC-helix. This structure-based classification has proved a better framework for categorizing mutations by TKI sensitivity and patient outcomes than previous exon-based grouping [27]. In this review, we primarily discuss the TKI sensitivity and selectivity for “classical” *EGFR* mutations and uncommon mutations based on structure and function studies.

## 2. The Conformational States of EGFR Kinase Domain

The EGFR kinase domain consists of the N-lobe and C-lobe with the ATP-binding site connecting the two lobes. The N-lobe is mainly formed of β-strands (β1-β5), the regulatory αC-helix and the glycine-rich phosphate-binding loop (P loop), whereas the larger C-lobe mainly consists of helices, with a conformationally variable activation loop (A loop). In the absence of ligand, the receptor predominantly exists as a monomer, with the kinase domains in inactive conformation [14]. In “Src-like inactive” conformation, the αC-helix is rotated outward (the “αC-out” conformation) and is maintained by a two-turn helix within the N-terminal portion of the A loop [16]. The catalytically important KE salt-bridge interaction between Lys745 and Glu762 is disrupted, while the Glu762-Lys860 salt bridge is formed [49]. Leu858 and Leu861 pack towards the αC helix, preventing the formation of the Lys745-Glu762 salt bridge (Figure 1A). In the presence of ligand, the receptors form dimers and multimers or higher-order oligomers, in which the kinase domains switch to active conformation [13]. In this conformation, the αC helix is rotated inwards the ATP-binding site (the “αC-in” conformation). The A loop maintains the β9 strand and forms an overall conformation well accommodating the substrate binding. The Lys745-Glu762 salt bridge is formed, and Leu858 and Leu861 are surface-exposed [16] (Figure 1B). In addition, during the transition from the active to “Src-like inactive” conformation, the intrinsically disordered conformation has been observed and featured the αC helix partially disordered and placed “out,” while the A loop maintains the β9 strand intact like in active conformation [21].

## 3. Targeted Therapy for NSCLC Patients with *EGFR* Mutations

Multiple targeted agents, including first-, second- and third-generation EGFR TKIs, have been approved or are under active investigation for patients with *EGFR*-mutant NSCLC (Table 1). The first-generation TKIs, including erlotinib and gefitinib, are reversible inhibitors, binding both to mutant as well as wild-type (WT) EGFR [9,10]. The second-generation TKIs, including afatinib and dacomitinib, are irreversible inhibitors that covalently bind to EGFR [28,50]. The third-generation EGFR TKIs, including osimertinib (AZD9291), aumolertinib (HS-10296) and alflutinib (AST2818), are irreversible inhibitors that selectively bind to mutant EGFR and show greater efficacy than the first- and second-generation TKIs [51]. Osimertinib has been approved by the FDA for both frontline and second-line treatment of NSCLC with *EGFR*-sensitizing mutations [52,53,54]. Aumolertinib (HS-10296) and alflutinib (AST2818) have been approved for the treatment of *EGFR*-mutant NSCLC in China [55,56,57]. Acquired resistance inevitably occurs, and a promising new generation of EGFR-targeting agents is under investigation [58,59].

### 3.1. Classical Mutations

We analyzed the GENIE lung cancer dataset (GENIE Cohort v12.0-public, n = 153,834) [60] and found that Ex19Del and L858R are found in 71.9% of NSCLCs with an *EGFR* mutation. Ex19Del and L858R are the sole EGFR mutations in 34.3% and 28.5%, respectively, while co-occurrence of Ex19Del or L858R with uncommon EGFR mutations accounts for 6.0% and 3.1% of cases (Figure 2a). Ex19Del and L858R as classical mutations confer sensitivity to all first-, second- and third-generation EGFR TKIs [10,20,22,55,56,61,62] (Figure 3), and co-occurrence of an uncommon mutation does not impact TKI sensitivity [40,63]. In fact, patients with complex “classical” mutations (Ex19Del or L858R in complex with an uncommon EGFR mutation) might have more favorable clinical outcomes than patients with uncommon mutations alone [64,65]. We have not found any classical + classical complex mutations in our analysis of GENIE dataset. The sensitivity of this class of complex mutations might be equivalent to the corresponding single mutation, as L858R + Ex19Del complex mutations display similar patient outcomes with EGFR TKI treatment compared to patients with single classical mutations alone [66].

Structurally, Leu858 lies within the two-turn helix of the A loop and packs against the αC helix in the “Src-like inactive” conformation. Arginine has a much larger side chain than leucine, and substitution of Leu858 with arginine is not tolerated with the inactive conformation [20]. The positively charged Arg858 is surrounded by a cluster of negatively charged residues (Glu758, Asp855, and Asp837), which stabilize the αC helix and the KE salt bridge. Therefore, the substitution holds the kinase domain in the active conformation likely at the expense of locally disordered conformation, rather than “Src-like inactive” conformation [21] (Figure 1). As a result, L858R mutation promotes the ligand-independent dimerization of EGFR kinase [24]. Structure-based approaches classify L858R into classical-like mutations, as Arg858 is located at the N-terminal portion of the A loop in the C-lobe, which is far from the ATP-binding pocket [27]. A spectrum of rare L858R + mutations were also categorized into classical-like mutations [27]. This subgroup of mutations is sensitive to all first-, second- and third-generation EGFR TKIs (Figure 4).

Ex19Del mutations shorten the length of β3-αC loop and suppress the locally disordered conformation [21], and therefore constrain the αC helix to the active conformation [22], with the KE salt bridge making the interaction between the kinase domain and ATP favorable [67]. There have been more than 20 Ex19Del variants identified in NSCLC [68]. The length of the β3–αC loop functions as a rheostat for kinase activity. Specifically, deletion of five amino acids (E746_A760) makes an optimal orientation of the αC helix for catalytic activity, while other deletions also result in structural perturbations of the αC helix that impair kinase activity compared to WT EGFR [22]. This conformational difference leads to variable ability to form ligand-independent dimerization and therefore differential sensitivity to EGFR TKIs [68,69]. Clinically, variations in Ex19Del are translated into differential clinical features and treatment outcomes [70]. Ex19Del mutations were classified into classical-like mutations as, structurally, the β3–αC loop in the N-lobe is far from the ATP-binding pocket [27]. A spectrum of Ex19Del + rare mutations were also categorized into classical-like mutations (Figure 4).

Ex19Del and L858R mutants have a lower competitive binding affinity for ATP than WT EGFR. First-generation EGFR TKIs bind more favorably to the kinase domain of mutants than WT, thereby inhibiting mutant kinase activities in a therapeutically useful manner [20]. The substitution of Thr790 to the bulky methionine increases the ATP affinity and stabilizes the active conformation of EGFR kinase domain, which sterically prevents the binding of first-generation TKIs, thereby imparting drug resistance [71]. Second-generation TKIs bind irreversibly to EGFR Cys797 at the margin of the ATP-binding cleft by covalent adduct formation, and were anticipated to overcome acquired resistance caused by T790M mutation [72]. However, the poor selectivity of second-generation TKIs for T790M over WT EGFR results in dose-limiting toxicities that limit clinical efficacy [61]. Osimertinib has a differential binding pose with Met790 from Thr790 of WT EGFR, thus yielding greater selectivity for T790M mutation over WT EGFR than afatinib [25]. Greater selectivity has translated into more favorable treatment outcomes, as either second-line or first-line therapy, than earlier-generation EGFR TKIs [52,53].

### 3.2. Uncommon EGFR Mutations

#### 3.2.1. Exon 18 Mutations

EGFR exon 18 mutations include E709_T710insX (e.g., exon 18 deletion, Ex18Del), E709X and G719X (Figure 2b). From our analysis of the GENIE dataset, either E709X or E709-T710insX as a single *EGFR* mutation accounts for around 0.68% of all *EGFR* mutations and 1.7% of rare *EGFR* mutations in NSCLC (data not shown). Around 96% of E709X co-occurs with another EGFR mutation, such as G719X or L861X (Appendix A). G719X mutations as independent *EGFR* mutations, including G719A, G719C, G719S, G719D and G719V, and account for 4.0% of rare *EGFR* mutations and 1.6% of all *EGFR* mutations in NSCLC. G719X mutations also co-occur with other *EGFR* mutations, which represent 8.4% of rare *EGFR* mutations and 3.2% of all *EGFR* mutations in NSCLC (Figure 2c).

Preclinical studies show that exon 18 mutations including E709K, E709_T710insD and G719A are sensitive to first-, second- and third-generation EGFR TKIs, but show less sensitivity than Ex19Del [39] (Figure 3). Exon18 mutations tend to be more sensitive to second-generation TKIs than first- and third-generation EGFR TKIs [40,68,69]. E709K and G719A tend to be more sensitive to EGFR TKIs than E709-T710insD [40,68]. Clinically, patients with G719X and Ex18Del respond poorly to first-generation TKIs [39], and patients with G719X also respond poorly to the third-generation TKI osimertinib [70]. Few clinical data about second- and third-generation TKIs exist for E709X as a single EGFR mutation.

Both Glu709 and Gly719 are located in the N-lobe. Glu709 is located on the N-terminal end of the β1 strand, which is immediately followed by the P loop. Gly719 is the first glycine in the evolutionarily conserved “GXGXXG” motif of the P loop. Substitution of Gly719 to non-glycine residues decreases the flexibility of the P loop and attenuates the hydrophobic interactions that constrain the αC-helix in the inactive conformation [20]. Therefore, G719X mutations destabilize the inactive conformation and are well accommodated in the active conformation [20,21]. This conformation increases the propensity for dimerization and subsequent receptor activation. Exon 18 mutations, including E709X, G719X and Ex18Del mutations, were categorized as PACC mutations [27] (Figure 4). Structurally, PACC mutations change the flexibility of the P loop and destabilize an osimertinib-binding pattern. By contrast, second-generation TKIs do not interact with the P loop and favor the binding pose. Therefore, PACC mutations are more sensitive to the second-generation TKIs than other-generation TKIs [27]. Clinically, patients with G719X mutations are more responsive to second-generation EGFR TKIs than first- and third-generation TKIs [42,72,73]. Similarly, patients who develop resistance to osimertinib via a mutation in Gly724, the third glycine in the “GXGXXG” motif, often respond to second-generation EGFR TKIs [74,75,76,77], but not first-generation TKIs [78]. The variable sensitivity of different exon 18 mutations observed in previous studies might be explained by the differential flexibility of the P loop induced by mutations.

#### 3.2.2. Exon 19 Mutations

We found 13 cases with L747P mutation in the GENIE dataset. We also found 5 cases with L747X complex mutations, including L747F + L861Q (1 case), L747S + G719C (2 cases) and L747A + I744M (2 cases). We have not found exon 19 insertions in the GENIE dataset, while this subset of mutations has been reported to account for approximately 1% of *EGFR*-mutant NSCLC [63]. Available clinical data indicate that L747P/S mutations are resistant to first- and third-generation EGFR TKIs [79,80] and confer sensitivity to second-generation EGFR TKIs [81,82]. Consistent with this evidence, L747X mutations were included in PACC mutations, as substitution of Leu747 to proline or serine might alter the rigidity of the β3–αC loop, which stabilizes the αC-helix and KE salt bridge interactions in the active conformation [83].

#### 3.2.3. Exon 20 Mutations

*EGFR* exon 20 insertion (Ex20ins) mutations contain a broad spectrum of small insertions of 1–7 amino acids that occur within the C-terminal end of the αC-helix and the immediately following loop [37]. After classical mutations, Ex20ins mutations are the next most prevalent *EGFR* mutation in NSCLC [84]. From the GENIE dataset, we found more than 60 different forms of Ex20ins mutations, accounting for 16% of rare *EGFR* mutations and 6.2% of all *EGFR* mutations in NSCLC (Figure 2). We also found other forms of exon 20 mutations, including S768I (3 cases), V774L (1 case) and Q787L (1 case) (Figure 3). Exon 20 mutations also co-occur with other *EGFR* mutations (Figure 2c).

Variable TKI sensitivity has been found in different types of Ex20ins mutations [85,86]. Most patients harboring Ex20ins mutations are insensitive to most of the FDA-approved EGFR TKIs [87,88,89,90]. However, a specific subset of insertions within the C-terminal portion of the αC-helix, for example, A763_Y764insFQEA, are sensitive to first-generation EGFR TKIs [44]. Multiple reports have demonstrated A763_Y764insFQEA is also responsive to second- and third-generation EGFR TKIs with comparable sensitivity to classical mutations [45,89,91]. For the rest of the Ex20ins mutations, most of them do not respond to first- and third-generation TKIs [41], but are sensitive to second-generation TKIs [92]. Exceptionally, the three insertion mutations (D770_N771insG, D770>GY, and N771_P772insN) are at least partially responsive to first-generation EGFR TKIs [41].

Structural variations in Ex20ins mutations lead to the heterogeneity of TKI sensitivity. The insertions form a wedge that “pushes” the αC-helix and hold the αC in the active conformation [23,44], leading to constitutive activation of the receptor [23,44]. Sequence and conformational variation in the αC–β4 loop may function as a rheostat that regulates the kinase activity via modulating the αC-helix conformation [23]. Exceptionally, the insertion A763_764insFQEA that occurs within the αC-helix differs from insertions within the αC–β4 loop, and may constitutively activate the receptor most similarly to classical mutations, rather than other Ex20ins mutations. As might be expected, A763_764insFQEA is sensitive to first-, second- and third-generation EGFR TKIs [45,89,91], and is classified as a classical-like mutation. Other Ex20ins mutations are categorized as a unique class called Ex20-L mutations [27]. Ex20ins-L are sensitive only to select second-generation TKIs such as poziotinib [93], and Ex20ins-active TKIs such as CLN-081 [94] and mobocertinib [95]. Ex20ins-L mutations could be further subdivided into Ex20ins near αC-β4 loop (Ex20ins-NL) and far αC–β4 loop insertions (Ex20ins-FL). Ex20ins-NL is more sensitive to second-generation and Ex20ins-active TKIs than Ex20ins-FL (Figure 4).

Patients with single S768I mutation have variable responses to first-generation EGFR TKIs [90,96,97,98]. Patients whose cancers harbor theS768I mutation responded to afatinib [87], leading to FDA approval of this drug for this subset of patients. Notably, S768I usually co-occurs with additional *EGFR* mutations (7/8 cases in this study), making the response of the single S768I mutation to afatinib ambiguous. Co-occurrence of S768I with additional *EGFR* mutations might impact TKI sensitivity [99,100]. One patient with S768I+V769L, for example, was resistant to afatinib [101]. S768I-complex mutations are sensitive to the third-generation EGFR TKI osimertinib [102,103], though little data exists on single-mutant S768I sensitivity. S768I and its complex mutations are classified as PACC mutations. Ser768 is located at the αC–β4 loop. Substitution of Ser768 to isoline improves the hydrophobic interactions between the αC helix and the adjacent β9 strand, which might enthalpically stabilize the “αC-in” active conformation [21]. As such, S768I and its complex mutations are classified as PACC mutations (Figure 4).

Finally, little clinical or structural data exist for V774L and Q787L, though V774M and its complex mutations are classified as PACC mutations.

#### 3.2.4. Exon 21 Mutations

Apart from L858R mutation, we found another exon 21 mutation L861X, including L861Q and L861R, in 2.3% of all *EGFR* mutations and 5.7% of rare *EGFR* mutations in the GENIE lung cancer dataset (Figure 2b). L861X mutations were also found to co-occur with additional *EGFR* mutations (Figure 2c). Preclinical and clinical studies show that L861Q mutation has an intermediate sensitivity to first-generation EGFR TKIs comparable to S768I and G719X mutations [42,43,104] (Figure 3). Patients with L861Q mutation are responsive to the second-generation EGFR TKI afatinib [87], which led to FDA approval of afatinib for this subtype [10]. In preclinical studies, L861Q mutations are sensitive to the third-generation EGFR TKI osimertinib [43], and a phase II trial reported patients with L861Q having partial responses to osimertinib [102]. Structurally, substitution of Leu861 to Gln allows the formation of new hydrogen bonds near the C-terminal of the αC helix that might enthalpically stabilize the “αC-in” active conformation [21]. L861X mutation and its complex mutations have been classified as PACC mutations (Figure 4).

#### 3.2.5. EGFR Kinase Domain Duplication

In-frame, tandem duplication of *EGFR* exons 18–25, encoding EGFR kinase domain duplication (EGFR-KDD), has been identified in patients with NSCLC [105], with cases of a duplication of exons 14–26, 17–25 and 18–26 also reported in NSCLC [106]. EGFR-KDD occurs in 0.2–0.24% of all *EGFR*-mutant NSCLC patients [106,107,108]. Current evidence suggests that EGFR-KDD is sensitive to multiple EGFR TKIs [105,106]. Case studies show that NSCLC patients with EGFR-KDD can have durable partial response to either first-line [106] or second-line treatment with gefitinib [109], though at least one patient with EGFR-KDD did to respond to first-line gefitinib [106]. Case studies also show that NSCLC patients with EGFR-KDD are at least partially responsive to afatinib treatment [105,110], and the third-generation TKI osimertinib has been found to be effective in patients with EGFR-KDD [111,112].

Structure–function studies indicate that EGFR-KDD can form ligand-independent intramolecular asymmetric dimers in which one kinase domain functions as “activator” (donor) and the other functions as “receiver” (acceptor), and ligand-dependent intermolecular asymmetric dimers and higher-order oligomer. The linker between the two kinase domains provides additional enthalpic stabilization to promote activation. Preclinical studies show that the inhibition of EGFR-KDD activity can be achieved by the inhibition of intramolecular kinase activity (afatinib) and intermolecular kinase activity (cetuximab) [48]. Compared to gefitinib, structural studies predict osimertinib to bind more favorably with, and thus better inhibit, EGFR-KDD [113]. More structure–function studies are required to understand the detailed conformations when EGFR-KDD binds to the available TKIs, so that we can classify this type of mutations based on the structure-based grouping.

#### 3.2.6. Uncommon Complex Mutations

We found uncommon complex mutations (rare + rare) occur in 36.5% of rare *EGFR* mutations and 6.7% of all *EGFR* mutations in the GENIE lung cancer dataset (Figure 2C). Of them, G719X complex mutations are the most common subtype of mutations and account for almost a half of all uncommon complex mutations (Figure 2C), with G719X + S768I most common one (Table 2). E709X complex mutations are the second most common of this subtype of mutations and account for 14.6% of all uncommon complex mutations (Figure 2C). Interestingly, all E709X mutations co-occur with G719X mutations (Table 2). We also found Ex20ins, L861X, L858X and S768I complex mutations (Figure 2C, Table 3).

Uncommon *EGFR* complex mutations encompass a wide range of patient responses to EGFR TKIs. Structure-based approaches classify the majority of uncommon EGFR complex mutations as PACC mutations, including E709X + G719X, G719X + rare and S768I + rare, like their single mutation partners. These complex mutations tend to yield more favorable patient outcomes in response to TKIs than the single rare mutations alone [88,104,114,115]. Interestingly, the sensitivity to EGFR TKIs is likely influenced by the specific co-occurring partner mutation. For example, patients with G719X + S768I mutations have dramatically different clinical outcomes compared to patients with G719X+L861Q mutations [104]. Single S768I mutation might not be sensitive to erlotinib, but co-occurrence with sensitizing *EGFR* mutations will confer sensitivity to the single S768I mutation [99].

## 4. Conclusions and Perspective

Structural studies on EGFR mutations have provided important information regarding EGFR biology and drug sensitivity. Comprehensive evaluation of structure alterations is very necessary for each mutations of EGFR kinase domain. Summary of these studies and the recent structure-based approaches have proposed a new classification of *EGFR* mutations. While many studies have emphasized the heterogeneity of a subset of mutations when grouped by exons, structure-based classification of *EGFR* mutations can better predict TKI sensitivity and patient outcomes than exon-based grouping. For example, one specific Ex20ins mutation, A763_764insFQEA, with similar activation mechanism to L858R and Ex19Del mutations, has been classified as a classical-like mutation, while the rest of the Ex20ins mutations have been classified as a unique group as Ex20ins-L mutations. This group of mutations have greater sensitivity to select second-generation EGFR TKIs than other groups.

Another advantage of the structure-based classification is identifying groups of mutations that, though occurring on different exons, confer similar sensitivity to EGFR TKIs. For example, G719X, L747P and S768I mutations, corresponding to exon 18, 19 and 20, respectively, were more sensitive to second-generation EGFR TKIs than first- and third-generation TKIs, and therefore were classified as PACC mutations. This kind of classification might provide important guidance for future oncology efforts, such as developing new TKIs, improving the design of clinical trials and matching patients with the best EGFR TKI.

Rare mutations, especially the complex mutations, need to be systematically evaluated by structure and function studies. Current studies mainly focus on common complex mutations. Nonetheless, preclinical and clinical data are limited. Preclinical studies are required to systematically analyze single rare mutations as well as the effects that complex partners add, both structurally and functionally. Rationally designed clinical studies, informed by structure–function research, will be required to generalize the relationship between mutations, TKI sensitivities, and patient outcomes.

## Figures and Tables

**Figure 1 biomolecules-13-00210-f001:**
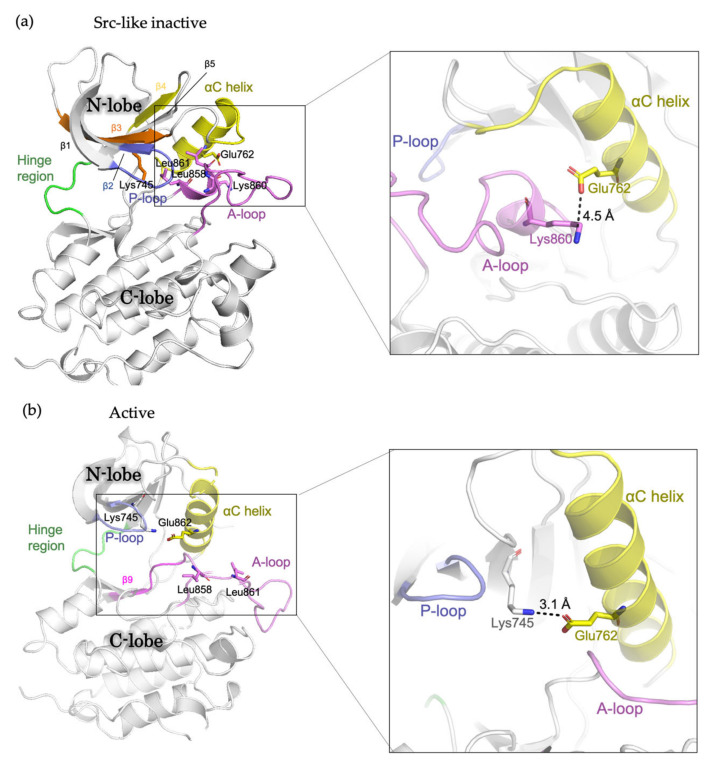
Schematic representation showing the structure of the EGFR kinase domain. (**a**) Src-like inactive conformation (PDB ID: 2GS7) and (**b**) active conformation (PDB ID: 2ITP) with important residues and structural features labelled.

**Figure 2 biomolecules-13-00210-f002:**
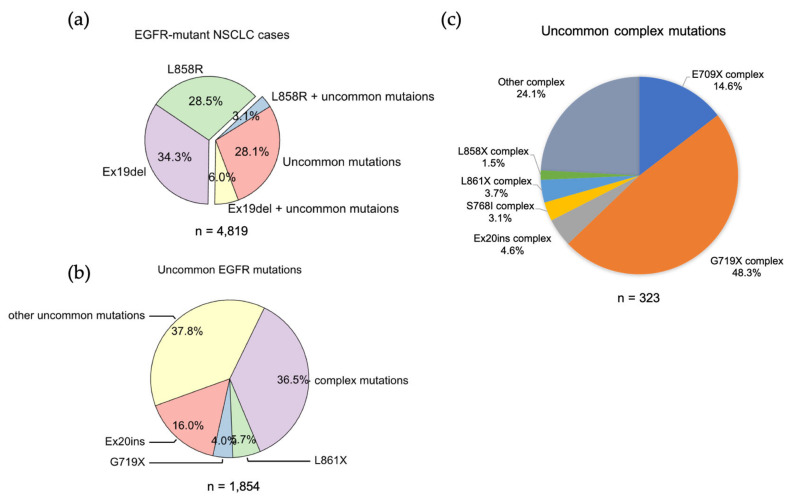
The frequencies of *EGFR* mutations in NSCLC. (**a**) Data were acquired from GENIE dataset (GENIE Cohort v12.0-public, n = 153,834). Data were filtered to contain mutations from NSCLC (n = 22,050). The common resistance mutations T790M and C797S were filtered out. (**b**) Pie chart showing the frequency of different uncommon EGFR mutations in NSCLC. (**c**) Pie chart showing the frequency of different uncommon complex EGFR mutations in NSCLC.

**Figure 3 biomolecules-13-00210-f003:**
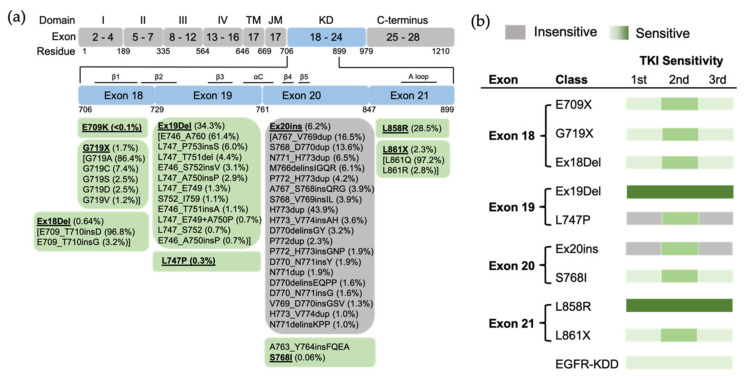
The position of *EGFR* mutations and sensitivity to EGFR TKIs based on exon-grouped classification. (**a**) Exons 18–21 in the tyrosine kinase region where the relevant mutations are located are expanded, and a detailed list of EGFR mutations in these exons is shown in the boxes below. TM, transmembrane; JM, juxtamembrane; KD, kinase domain. (**b**) The sensitivity to EGFR TKIs based on exon-grouped classification. Gray, insensitive to EGFR TKIs. Green, sensitive to EGFR TKIs in which the light-to-strong color represents weak-to-strong sensitivity.

**Figure 4 biomolecules-13-00210-f004:**
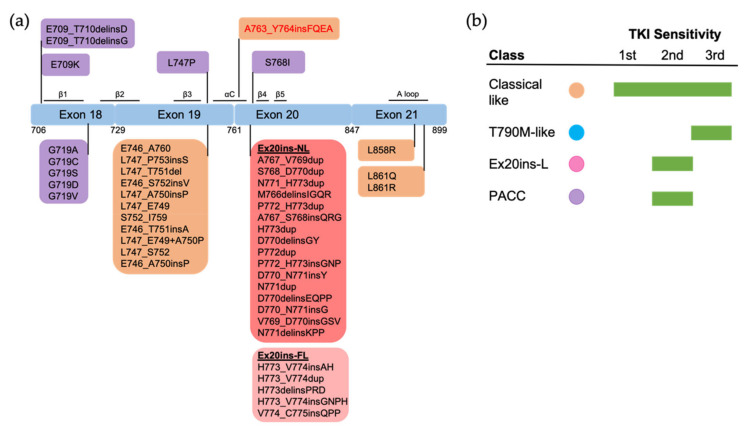
The positions and structure-based classification of *EGFR* mutations. (**a**) A detailed list of *EGFR* mutations in each exon are shown in the below boxes. Different colors of each box represent different classes of mutations following structure-based classification. Ex20ins-L, exon 20 loop insertions; Ex20ins-NL, exon 20 near-loop insertions; Ex20ins-FL, exon 20 far-loop insertions; PACC, P-loop and αC-helix compressing mutations. (**b**) The sensitivity to EGFR TKIs of each group of mutations based on structure-based classification.

**Table 1 biomolecules-13-00210-t001:** EGFR TKIs currently in clinical use or in development for patients with *EGFR*-mutant NSCLC.

EGFR TKI		Selectivity	Binding Mode	Status in NSCLC
1st-generation	Erlotinib	WT EGFR	Reversible	FDA-approved
	Gefitinib	WT EGFR	Reversible	FDA-approved
2nd-generation	Afatinib	WT EGFR	Irreversible	FDA-approved
	Dacomitinib	WT EGFR	Irreversible	FDA-approved
3rd-generation	Osimertinib (AZD9291)	Mutant EGFR	Irreversible	FDA-approved
	Aumolertinib (HS-10296)	Mutant EGFR	Irreversible	Approved in China [55]
	Alflutinib (AST2818)	Mutant EGFR	Irreversible	Approved in China [56]
4th-generation	JBJ-09-063	Mutant EGFR	Reversible	Under investigation [59]
	BLU-945	Mutant EGFR	Reversible	Under investigation [58]

**Table 2 biomolecules-13-00210-t002:** Complex mutations identified in exon 18 from GENIE lung cancer dataset.

E709X	Partner Mutation	Count	G719X	Partner Mutation	Count
E709A	G719A	12	G719C	S768I	43
E709K	G719A	11	G719A	S768I	24
E709A	G719R	8	G719S	S768I	14
E709A	G719C	3	G719A	L861Q	12
E709K	G719S	3	G719S	L861Q	11
E709K	G719C	2	G719A	R776H	6
E709V	G719C	2	G719A	L833V	5
E709V	G719S	1	G719S	S768N	3
E709A	L861R	1	G719S	R776H	3
E709A	L861Q	1	G719D	L861Q	2

**Table 3 biomolecules-13-00210-t003:** Complex mutations identified in exon 20 and 21 from GENIE lung cancer dataset.

Ex20ins	Partner Mutation	Count	L861X	Partner Mutation	Count
N771_H773dup	V845L	2	L861Q	S720F	3
D770delinsEQPL	H773Y	1	L861Q	S768I	2
D770delinsEL	N771Y	1	L861R	S768I	1
D770_V774dup	Q791E	1	L861R	P1073T	1
D770_P772dup	H773Y	1	L861Q	R776H	1
D770_N771insY	S1030L	1	L861Q	R776C	1
D770_N771insG	N771Y	1	L861Q	L838V	1
A767_V769dup	R836C	1	L861Q	L833F	1
A767_S768insQRG	V765L	1	L861Q	L747F	1
H773delinsQI	N771H	1			

## Data Availability

The authors declare that all data supporting the findings of this study are provided in the Appendix A file. The GENIE Cohort v12.0-public dataset is publicly available through Sage Bionetworks (https://www.aacr.org/professionals/research/aacr-project-genie/aacr-project-genie-data/, accessed on 10 January 2023).

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
