# Peer review of "Structure-Guided Strategies of Targeted Therapies for Patients with EGFR-Mutant Non–Small Cell Lung Cancer"

_biomolecules, 2023, doi:10.3390/biom13020210_

Round 1

Reviewer 1 Report

In the review article, the authors attempted to summarize studies on the structure-based classifications of EGFR mutations, mechanisms, and drug response/sensitivity in EGFR-mutant NSCLC.

The topic is narrow and straightforward.  The authors also just gave simple summary of the different EGFR mutations and the treatment responses, and brief explanations with regards to mechanisms.

The following are some of the specific corrections and suggestions before it can be accepted:

Lines 23-24: “In this review, we will summarize the recent progress on targeted therapy strategies for patients with EGFR-mutant NSCLC.”

This statement is too general and did not clearly state the objective of the review article.

Lines 112-113: "are reversibly inhibitors binding to"

change to 'are reversible inhibitors that bind to'

Figure 3

Put labels on the 2 subfigures (left and right). Indicate the meaning of the color in terms of drug sensitivity.

Figures 3 and 4 

Figures 3 and 4 titles are the same. Either write the distinct title for each figures, or combine the figures into one and label subfigures a and b accordingly. 

Figure 5

The figures are not clear, you may tabulate instead in order to immediately understand what is being described.

The addition of figures that visually help the reader to focus on what is described in the text as well as tables that summarize what were mentioned and described in the text is highly recommended.

Reviewer 2 Report

Minor points

1. Some grammatical errors are found in the manuscript. The manuscript seems to need English proofreading.

2. A few sentences need citations (page 1, line 31; page 10, lines 366, 367, and 372).

3. The authors should explain about A763_Y764insFQEA in more detail (page 2, line 63).

4. The authors describe the strcuture of the EGFR kinase domain by indicating specific secondary structures such as beta1-10. However, we cannot find the structural inforamtion from Figure 1. Thus, the authors should prepare an overall structure of the EGFR kinase domain, if the  secondary structures are inidcated in the manuscript.

5. The authors state that the two Lys residues form a salt bridge (page 2, line 89). Please explain how the two positively charged residues can form a salt bridge, and what distance between the two residues is.

6. Which residue interacts with Lys745 to form a salt bridge, Glu762 or Glu752. The sentence says Glu762, while Figure 1 does Glu752.

7. The title of Figure 1 could be more concise.

8. The section number in 3. Perspective must be changed into 4.
